# Alternative Approach for Specific Tyrosinase Inhibitor Screening: Uncompetitive Inhibition of Tyrosinase by *Moringa oleifera*

**DOI:** 10.3390/molecules26154576

**Published:** 2021-07-29

**Authors:** Farah J. Hashim, Sukanda Vichitphan, Jaehong Han, Kanit Vichitphan

**Affiliations:** 1Graduate School, Khon Kaen University, Khon Kaen 40002, Thailand; farah_bio_2020@yahoo.com; 2Department of Biotechnology, Faculty of Technology, Khon Kaen University, Khon Kaen 40002, Thailand; sukanda@kku.ac.th; 3Department of Biology, College of Science, University of Baghdad, Baghdad 10071, Iraq; 4Fermentation Research Center for Value Added Agricultural Products (FerVAAP), Khon Kaen University, Khon Kaen 40002, Thailand; 5Metalloenzyme Research Group and Department of Plant Science and Technology, Chung-Ang University, Anseong 17546, Korea

**Keywords:** inhibition kinetics, luteolin, *Moringa oleifera*, screening, tyrosinase, uncompetitive inhibitor

## Abstract

Tyrosinase (TYR) is a type III copper oxidase present in fungi, plants and animals. The inhibitor of human TYR plays a vital role in pharmaceutical and cosmetic fields by preventing synthesis of melanin in the skin. To search for an effective TYR inhibitor from various plant extracts, a kinetic study of TYR inhibition was performed with mushroom TYR. Among *Panax ginseng*, *Alpinia galanga*, *Vitis vinifera* and *Moringa oleifera*, the extracts of *V. vinifera* seed, *A. galanga* rhizome and *M. oleifera* leaf reversibly inhibited TYR diphenolase activity with IC_50_ values of 94.8 ± 0.2 µg/mL, 105.4 ± 0.2 µg/mL and 121.3 ± 0.4 µg/mL, respectively. Under the same conditions, the IC_50_ values of the representative TYR inhibitors of ascorbic acid and kojic acid were found at 235.7 ± 1.0 and 192.3 ± 0.4 µg/mL, respectively. An inhibition kinetics study demonstrated mixed-type inhibition of TYR diphenolase by *A. galanga* and *V. vinifera*, whereas a rare uncompetitive inhibition pattern was found from *M. oleifera* with an inhibition constant of *K_ii_* 73 µg/mL. Phytochemical investigation by HPLC-MS proposed luteolin as a specific TYR diphenolase ES complex inhibitor, which was confirmed by the inhibition kinetics of luteolin. The results clearly showed that studying TYR inhibition kinetics with plant extract mixtures can be utilized for the screening of specific TYR inhibitors.

## 1. Introduction

Treatment of skin disorders such as melasma, freckles, dark spots and hyperpigmentation is of great importance in the development of pharmaceutical and cosmetic products. The main cause of these problems is due to the accumulation of large amounts of melanin, produced by melanogenesis of melanosomes in the keratinocytes of the epidermis. Although melanogenesis occurs as a protective mechanism of the skin from UV exposure, unusual melanin stimulation produces free radicals causing wrinkles and damage to skin cell integrity [1]. Tyrosinase (TYR) is known as a key enzyme in the controlling step of the melanogenesis pathway [2]. The redox active type III copper-containing enzyme, belonging to the polyphenol oxidases, catalyzes two consecutive reactions of monooxygenase and oxidase [3]. The first reaction is known as monophenolase because l-tyrosine is oxygenated to l-dihydroxyphenylalanine (l-DOPA). The second reaction, diphenolase, oxidizes l-DOPA to l-dopaquinone (Figure 1) [4]. The catalytic study of TYR has been studied with various substrates to measure monophenolase and diphenolase activities [5]. In this study, l-DOPA was used as the substrate to investigate the inhibition kinetics of various plant extracts during dopachrome production.

While synthetic TYR inhibitors developed based on the catalytic mechanism can exhibit strong inhibition, these compounds also often establish cytotoxic and mutagenic effects against mammalian cells [6]. At the same time, most TYR inhibitors exhibiting melanin biosynthesis of melanocytes fail clinical trials because of toxicity and low activity. This is in part due to non-specific inhibition by the TYR inhibitors. Accordingly, more specific inhibitors need to be discovered, such as uncompetitive inhibitors that act on the TYR·l-DOPA complex. In the case of rare uncompetitive inhibition, the inhibitor binds only to the ES complex resulting in lower apparent *K_m_* (Michaelis constant) and *V_max_* (maximum reaction rate) when compared to the TYR activity without an inhibitor, and a high concentration of substrate cannot overcome the inhibition [7,8,9].

Inhibitors from natural resources, such as plant extracts, often suffer from low and non-specific inhibition. Besides, the vagueness of the inhibition mechanism of the extracts greatly hinders practical application, regardless of many advantages, such as low cost, direct production with less formulation, and safety [10]. Regardless, searching the potential TYR inhibitors from natural resources with low toxicity is continuously performed even with a low success rate, and alternative methods making up for such a drawback are still sought [2].

Therefore, we have investigated the inhibition of TYR by selected plant extracts by adopting a TYR inhibition kinetic study. In detail, high phenolic plants consumed by Asian people in their daily diet, including *Panax ginseng* (ginseng rhizomes and leaves), *Alpinia galanga* (galanga rhizomes and leaves) *Vitis vinifera* (grape seeds) and *Moringa oleifera* (moringa leaves), were selected. Along with total phenolic contents, TYR inhibition and inhibition kinetics were studied to determine the TYR inhibition type, and the bioactive compound in the extract of the uncompetitive inhibitor was further investigated by HPLC-MS and detailed inhibition kinetics.

## 2. Results

### 2.1. TPC and TYR Inhibition

Polyphenolics are a group of plant secondary metabolites synthesized by shikimate and polyketide pathways. Many of these compounds with catechol or resorcinol functional groups, such as flavonoids, were known to inhibit TYR activity [11], and are resistant to intestinal metabolism by gut microbiota [12,13,14]. In this study, 70% ethanol was used to enrich the polyphenolic compounds from the plant samples. In the case of *A. galanga*, 95% ethanol was instead used, due to the high concentration of hydrophobic composition. Table 1 shows TPC of each extract expressed in mg of GAE/g of dry weight extract. The highest concentration of phenolic content was found in *A. galanga* rhizome BL (651.9 ± 4.2 mg), followed by *M. oleifera* leaf (472.4 ± 2.8 mg) and *A. galanga* leaf (448.1 ± 2.3 mg). *P. ginseng* leaf showed lowest phenolic content among the tested plants.

All plant extracts inhibited TYR diphenolase activity in a dose-dependent manner, and the results were reported in terms of IC_50_ (Table 1). The extracts of *A. galanga* rhizome BL, *M. oleifera* leaf, and *V. vinifera* seed showed strong inhibition on TYR diphenolase, and the IC_50_ values were found at 105.4 ± 0.2 µg/mL, 121.3 ± 0.4 µg/mL, and 94.8 ± 0.2 µg/mL, respectively. They showed stronger TYR diphenolase inhibition than ascorbic acid and kojic acid, of which the IC_50_ were 235.7 ± 1.0 µg/mL and 192.3 ± 0.4 µg/mL, respectively. Strong TYR inhibition by *A. galanga* extract seems to be due to the high contents of curcuminoids, because curcuminoids inhibit proliferation of human melanocytes [15]. Additionally, industrial applications of grape seeds have recently been increasing due to the high content of health-promoting phytochemicals [16]; this reported TYR inhibition would further valorize its applications.

The correlation between TPC and IC_50_ of TYR inhibition was determined by Pearson coefficient (bivariate) correlation. The result showed moderate correlation (*r* = −0.79, *p* < 0.05), which means there is a good negative correlation between TPC and IC_50_ values (Figure 2). The result implies that the inhibition on TYR diphenolase activity could be mainly due to the polyphenolic compounds in the extracts.

### 2.2. Kinetics Study of TYR Inhibition by A. galanga BL, M. oleifera and V. vinifera

The extracts of the *A. galanga* rhizome BL, *M. oleifera* leaf and *V. vinifera* seed with low IC_50_ values were selected for the kinetics study of TYR inhibition. Catalytic studies using l-DOPA substrate are known to follow the Michaelis–Menten model under aerobic conditions [5]. Kinetic parameters of *K_m_* and *V_max_* were determined as 0.41 ± 0.20 mg/mL (2.0 mM) and 1.01 ± 0.24 OD_492_/min, respectively, for the diphenolase activity of mushroom TYR used for the experiments. The *K_m_* value obtained from our assay system was similar to the reported value [17]. The slopes and *y*-intercepts in the Lineweaver–Burk plot obtained in the presence of various concentrations of the inhibitor were used to determine the inhibition constants of *K_i_* and *K_ii_* (Table 2).

The inhibition kinetics of *A. galanga* rhizome BL on TYR diphenolase activity at different concentrations of the extract exhibited mixed-type inhibition as shown in Figure 3A. The Lineweaver–Burk plots at different concentrations of the inhibitors did not intersect on either *x* or *y* axis. The inhibition constants of *A. galanga* zhizome BL were determined from the secondary plots (Figure 3B,C). The intercept on the *x*-axis in Figure 3B, plotted between the inhibitor concentration and the slope of each line in Figure 3A, determined *K_i_* as 200 µg/mL. Similarly, the inhibition constant of *K_ii_* was determined as 128 µg/mL from Figure 3C, plotted between the inhibitor concentration and the intercept of each line in Figure 3A. Based on the inhibition constants, TYR diphenolase inhibition by *A. galanga* rhizome was stronger for the ES complex. Likewise, the inhibition by *V. vinifera* seed was also found as mixed-type (Figure 4). Inhibition constants of *K_i_* and *K_ii_* were determined as 42 µg/mL and 68 µg/mL, respectively. Strong inhibition by the extract of *V. vinifera* seeds may require further investigation. However, the mixed-type inhibition by both extracts was not unexpected, considering the complex effects on TYR by the various compounds in the plant extracts. 

Of significance, uncompetitive inhibition on TYR diphenolase was found from the *M. aleifera* leaf as shown at Figure 5. The Lineweaver–Burk plots at Figure 5A showed parallel lines, and *K_ii_* value of 73 µg/mL was determined from the secondary plot (Figure 5B). Considering the potent synthetic uncompetitive TYR inhibitors that were reported within a few µM of *K_ii_*, *M. aleifera* leaf is considered a very strong natural TYR inhibitor [18]. Uncompetitive inhibition is rare because it does not interact with the free enzyme, but only inhibits the ES complex. The uncompetitive inhibition and very low *K_ii_* value determined from the extract of *M. oleifera* leaf strongly suggested the presence of a potent TYR diphenolase inhibitor in the mixture, which specifically inhibited the ES complex. 

### 2.3. Compositional Analysis of M. oleifera Leaf Extract

The phytochemical composition of the *M. oleifera* leaf was investigated to search the active components responsible for the strong uncompetitive inhibition of TYR diphenolase. Among the major compounds identified from HPLC-MS analysis, (Figure 6, Appendix A), luteolin and its glucosides, luteolin 7-methyl glucuronide and luteolin 7-(6‴-acetylallosyl-(1→2)-glucoside), drew out attention, because luteolin was identified as a major component of the extract (1.6%, Appendix A) and was reported as an uncompetitive TYR inhibitor [19]. In order to confirm the unique uncompetitive inhibition property of *M. oleifera* extract, the TYR inhibitory kinetics of luteolin was further investigated (Figure 7). The inhibition constant of *K_ii_* was found to be 33 µg/mL (113 μM) for the inhibition of TYR diphenolase activity by luteolin (Table 2). The determined inhibition constant *K_ii_* was comparable to the reported value of 103 μM [19], but lower than that of *M. oleifera* leaf extract (73 µg/mL). Apparently, the extract of *M. oleifera* contains other bioactive compounds that may interfere with TYR catalysis. Furthermore, the extract contained only 1.6% of luteolin. Therefore, we propose the glycosides of luteolin, such as luteolin 7-methyl glucuronide and luteolin 7-(6‴-acetylallosyl-(1→2)-glucoside), can also be uncompetitive TYR diphenolase inhibitors, based on the HPLC-MS analysis.

## 3. Discussion

Activity-guided isolation of bioactive compounds from natural resources has been a popular method for the development of value-added products, including medicines, functional foods, and cosmetics. In this study, we aimed to search an alternative methodology in the screening of TYR diphenolase inhibitors from the widely consumed plant extracts. Aas long as effective uncompetitive inhibition is screened, investigation of the enzyme inhibition kinetics of plant mixtures can be an efficient approach for the development of specific TYR inhibitors due to the guaranteed safety of plant extracts. 

The results showed that *M. oleifera*, *V. vinifera* and *A. galanga* exhibited strong inhibition against TYR diphenolase, even compared with ascorbic acid and kojic acid, which are renowned TYR inhibitors. Further inhibition kinetics investigation was applied to elucidate the inhibition mechanism of the plant extracts. The observed mixed-type inhibition by *V. vinifera* and *A. galanga* implied that the compounds in the extract act non-specifically on both the free enzyme and ES complex at low concentrations. Interestingly, the uncompetitive modes of inhibition by *M. oleifera* extract strongly suggested specific ES complex inhibition by unidentified bioactive compounds. HPLC-MS analysis provided the phytochemical composition of *M. oleifera,* and luteolin and its glycosides were proposed as uncompetitive inhibitors. The inhibition by luteolin and *M. oleifera* in the presence of non-ionic detergent, triton X-100, was also performed to confirm that these are not Pan-Assay Interfering Substances (PAINS) [20]. The result has confirmed that they are specific uncompetitive inhibitors of TYR diphenolase (Appendix A). The rate-limiting step of TYR diphenolase was known to be the reaction step between l-DOPA and oxy-TYR [3]; hence, it appears luteolin and its glycosides inhibit the oxy-TYR·l-DOPA complex during the catalytic reaction. 

## 4. Materials and Methods

### 4.1. Chemicals and Reagents

Dimethyl sulfoxide (DMSO), Folin–Ciocalteau reagent, and gallic acid were purchased from Fluka, Switzerland. Mushroom tyrosinase (2587 units/mg), luteolin, and l-dihydroxylphenylalanine (l-DOPA) were purchased from Sigma-Aldrich, St. Louis, MO, USA.

### 4.2. Plant Materials and Extraction

Leaves and rhizomes of *M. oleifera*, *A. galanga* and *P. ginseng* were collected from local gardens in Khon Kaen, Thailand, during October 2017, and *V. vinifera* seeds were provided by Visootha (Nikki) Lohinavy, Gran Monte Asoke Valley in Pak Chong, Nakorn Rachasima, Thailand. Clean plant materials were dried by hot air oven at 65 °C (FD240, Binder, Germany) and powdered. All plant samples were extracted with 70% ethanol in a 1:4 *w/v* ratio, except *A. galanga* rhizomes which was extracted by 95% ethanol in the same ratio of sample to solvent. The mixture was stirred for 8 h at room temperature, and then filtrated by Whatman filter paper no. 1. All extracts were concentrated by rotary evaporator (Heidolph, VV2000, Schwabach, Germany). *A. galanga* rhizomes extract resulted in two layers, an oily yellow upper layer (YL) and brown lower layer (BL). All extracts were stored at 2 °C.

### 4.3. Total Phenolic Content

Total phenolic content (TPC) was measured by the Folin–Ciocalteu method [21]. Gallic acid with different concentrations of 50, 150, 250, and 500 mg/L was used for the construction of the standard curve. The reaction mixture was prepared by mixing 20 µL of sample (10 mg/mL in DMSO) with 1.58 mL of deionized water and 100 µL of Folin–Ciocalteu reagent. After incubating the reaction mixture at 25 °C for 5 min, 300 µL of 20% sodium carbonate was added. The absorbance was determined by spectrophotometer (2501PC, Shimadzu, Japan) at 765 nm. The TPC concentration was expressed as milligrams of gallic acid equivalent (GAE) per gram of dry extract.

Pearson coefficient (bivariate) correlation was determined by using the following equation,
$$r=\frac{\sum \left(xi-\bar{x}\right)\left(yi-\bar{y}\right)}{\sqrt{\sum {\left(xi-\bar{x}\right)}^{2}{\left(yi-\bar{y}\right)}^{2}}}$$
where *r* = correlation coefficient, *xi* = TPC, $\bar{x}$ = mean of *xi*, *yi* = IC_50_, and $\bar{y}$ = mean of *yi*.

### 4.4. Tyrosinase Inhibition

TYR inhibition was detected by the inhibition of the l-DOPA conversion into the colored dopachrome according to the published method [22]. Briefly, different concentrations of each plant extract were added into 96-well plates with 50 µL of mushroom TYR (300 U/mL in 0.1 mM phosphate buffer (PBS), pH 6.8), and the volume was adjusted to 200 µL with PBS (pH 6.8). The reaction mixture was incubated at 37 °C for 30 min. After that, 40 µl of l-DOPA (15 mM in 0.1 M PBS pH 6.8) was added to each well and incubated at 37 °C for 30 min in a dark place. Absorbance was recorded by microplate reader spectrophotometer (Imark, Biorad) at 492 nm. Kinetic parameters of *K_m_* and *V_max_* for TYR diphenolase activity with l-DOPA were determined by using the Lineweaver–Burk equation:$$\frac{1}{v}=\frac{K_{M}}{V_{max}}\frac{1}{\left[S\right]}+\frac{1}{V_{max}}$$

The concentration of plant extract which inhibited 50% enzyme activity was determined as IC_50_. All studies were conducted in triplicates and compared statistically. Kojic acid and ascorbic acid were used as the reference for TYR inhibitors.

### 4.5. Kinetic Study of TYR Inhibition

The extracts of *A. galanga* rhizome BL (50, 100 and 150 µg/mL), *V. vinifera* (25, 50 and 100 µg/mL), *M. oleifera* (125, 250 and 375 µg/mL), as well as luteolin (50, 75 and 100 µg/mL), were selected to determine the type of inhibition by Lineweaver–Burk plots. Each extract was added to the reaction mixture of 0.5–5.0 mg/mL of l-DOPA and 100 μL of 0.1 mM PBS (pH 6.8). The reaction was initiated by adding 50 μL of mushroom TYR at 37 °C. After incubation for 30 min, the absorbance was recorded by microplate reader spectrophotometer at 492 nm. The kinetic parameters of *K_m_* and *V_max_* were assessed from the original simulation dataset, using a Lineweaver–Burk plot of data obtained from different l-DOPA concentrations. The inhibition constant of *K_i_* (inhibition constant for free enzyme) was determined from the concentrations of the extract and the slopes of the Lineweaver–Burk plot by using the equation: $slope=\frac{K_{M}}{V_{max}}\left(1+\frac{\left[I\right\}}{K_{i}}\right)$, and *K_ii_* (inhibition constant for ES complex) was determined from the concentrations of the extract and the *y*-intercepts of the Lineweaver–Burk plot by using the equation: $intercept=\frac{1}{V_{max}}\left(1+\frac{\left[I\right\}}{K_{ii}}\right)$.

### 4.6. HPLC-MS Analysis of M. oleifera Extract

The extract of *M. oleifera* leaf (100 mg/mL) in 50% MeOH was ultra-sonicated for 20 min and centrifuged (10,000 rpm at 4 °C) for 10 min. The supernatant was filtrated through the 0.2 µm nylon membrane and the filtrate was injected into a liquid chromatograph-quadrupole time-of-flight mass spectrometer (LC-QTOF MS), (1290 Infinity II LC-6545 Quadrupole-TOF, Agilent Technologies, Santa Clara, CA, USA). HPLC was equipped with a Zorbax Eclipse Plus column (C18 2.1 × 150 mm, 1.8 µ, Agilent Technologies, Santa Clara, CA, USA). Formic acid (0.1%) in water (solvent A) and acetonitrile (solvent B) were used as the mobile phase and the gradient elution was as follows: 98% A, 0–2 min; 90% A, 2–25 min; 85% A, 25–40 min; 80% A, 40–48 min; 75% A, 48-68 min; 70% A, 68–80 min; 50% A, 80–85 min; 0% A, 85–90 min.; 98% A, 90–100 min. Peaks were detected at the wavelengths of 254 nm and 280 nm. The MS spectra were acquired in both positive and negative ion modes with auto MS/MS. Full-scan mode from *m*/*z* 100 to 1700 was applied. The HPLC peaks were identified by using the spectrum database for organic compounds in the METLIN database.

## Figures and Tables

**Figure 1 molecules-26-04576-f001:**
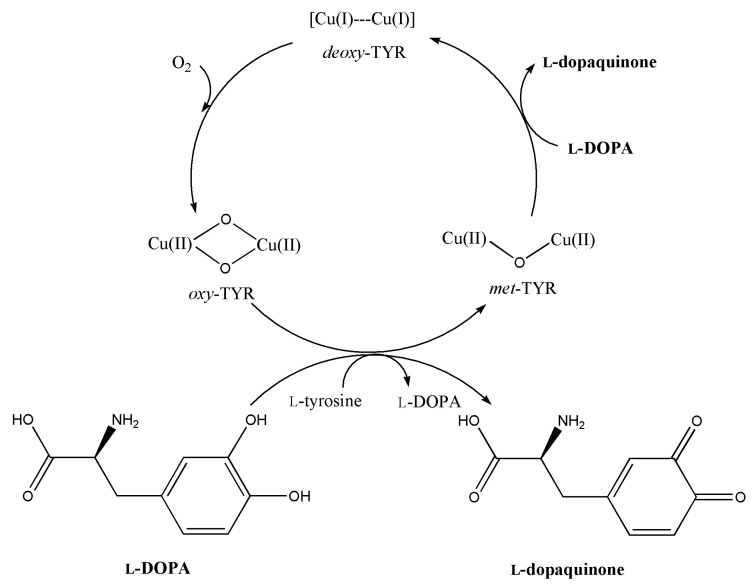
Schematic presentation of TYR catalysis on l-tyrosine and l-DOPA. *Deoxy*-TYR readily binds O_2_ to form *oxy*-TYR, which can act as monophenolase or diphenolase depending on the substrate. The *met*-TYR can act only as diphenolase.

**Figure 2 molecules-26-04576-f002:**
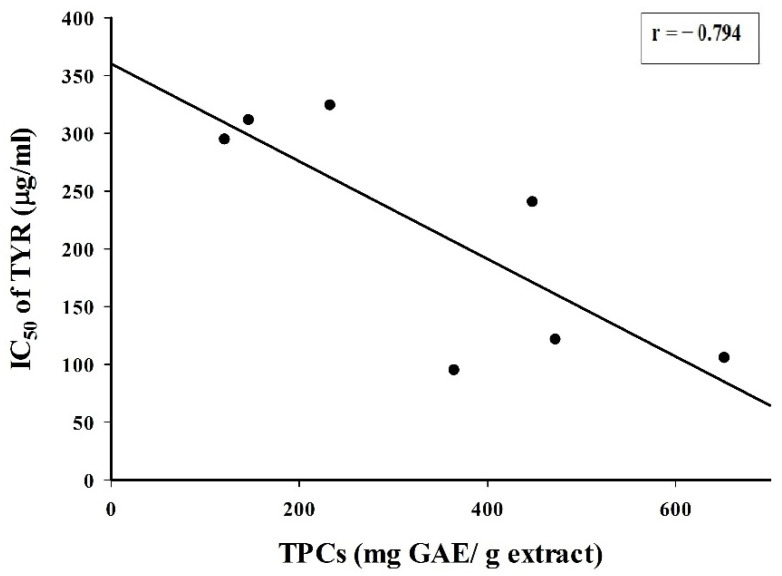
The correlation between TPC and IC_50_ of TYR inhibition of plant extracts (r = −0.78, *p* < 0.05).

**Figure 3 molecules-26-04576-f003:**
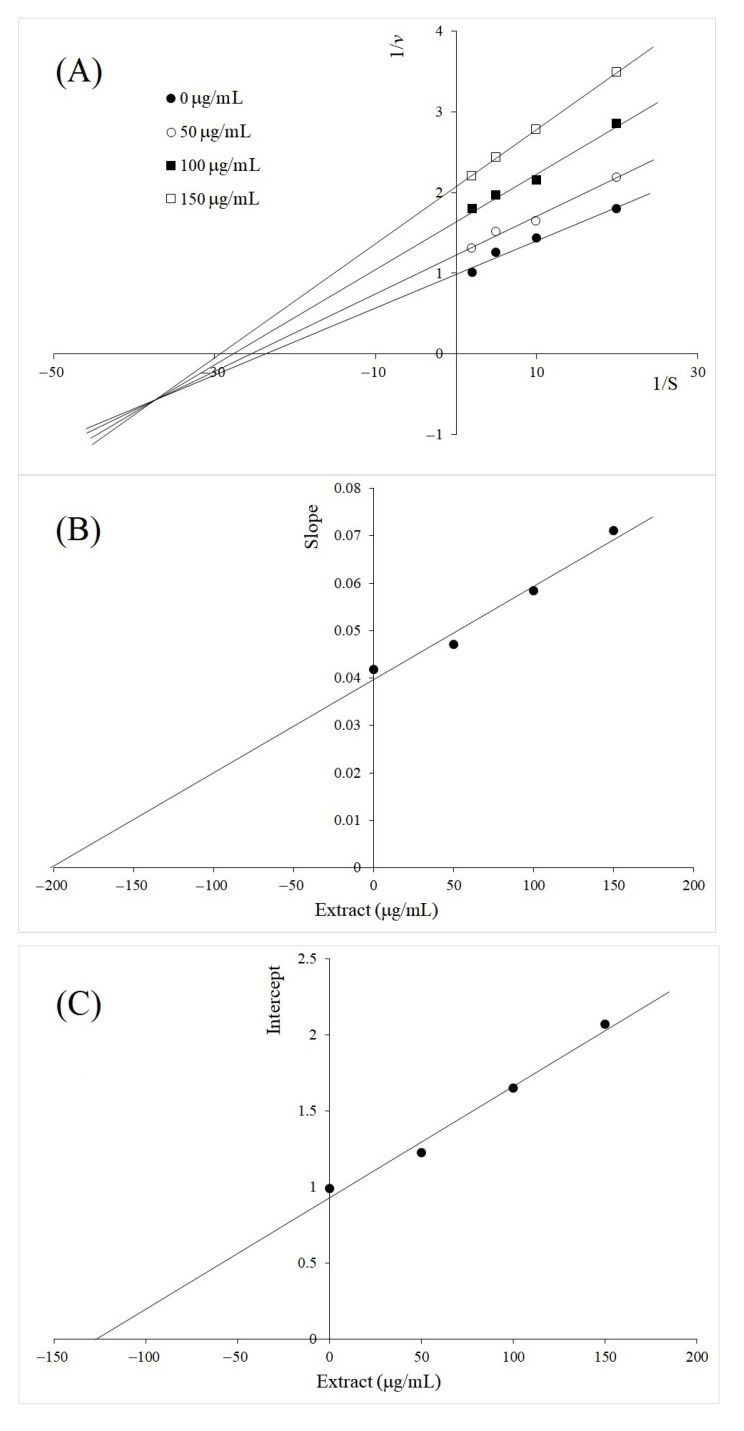
Inhibition kinetics of *Alpinia galanga* rhizome BL on TYR diphenolase activity at different concentrations (50, 100 and 150 µg/mL). (**A**) Lineweaver–Burk plot; (**B**) secondary plot between inhibitor concentration and the slope (linear equation; y = 0.0001980x + 0.03970, *R*^2^ = 0.9704); (**C**) secondary plot between inhibitor concentration and intercept (linear equation; y = 0.007311x + 0.9331, *R*^2^ = 0.9856).

**Figure 4 molecules-26-04576-f004:**
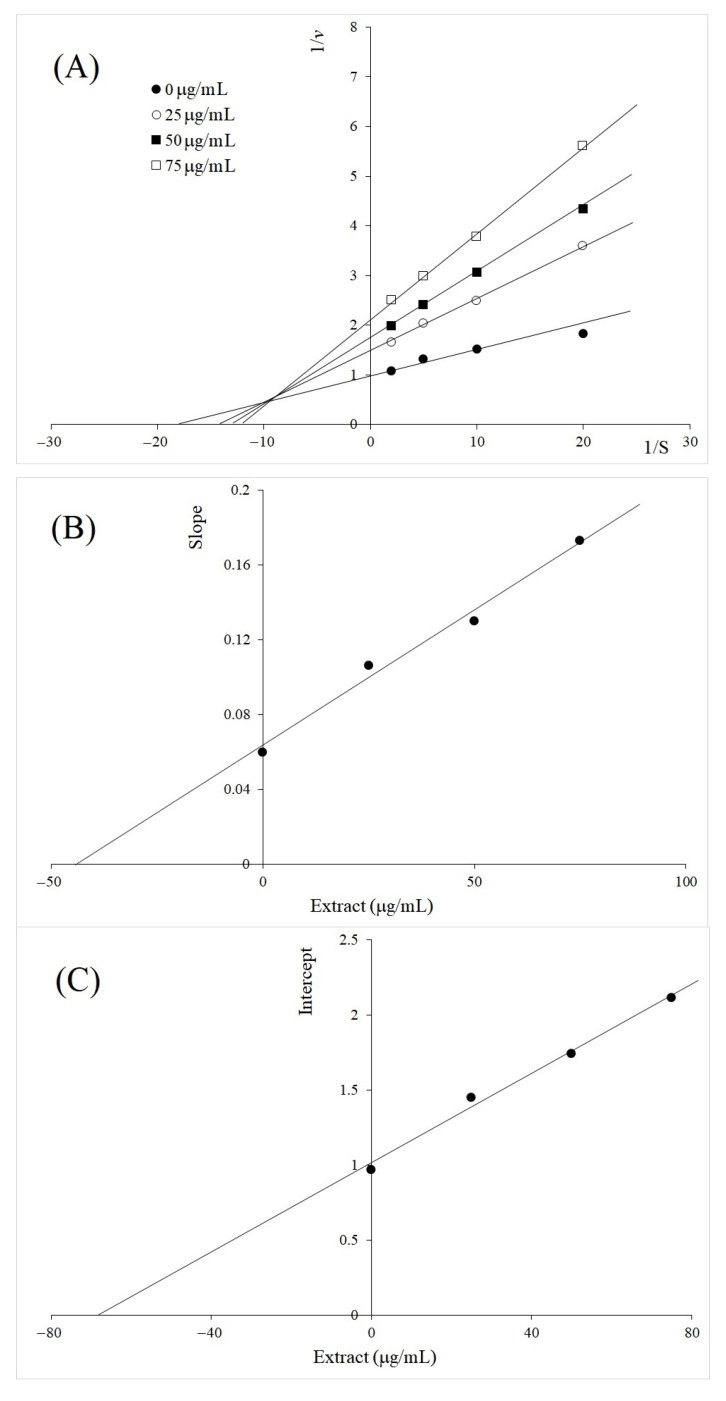
Inhibition kinetics of *V. vinifera* on TYR diphenolase activity at different concentrations (25, 50, and 100 µg/mL). (**A**) Lineweaver–Burk plot; (**B**) secondary plot between inhibitor concentration and the slope (linear equation; y = 0.001452x + 0.06289, *R*^2^ = 0.9868); (**C**) secondary plot between inhibitor concentrations and intercept (linear equation; y = 0.01492x + 1.011, *R*^2^ = 0.9903).

**Figure 5 molecules-26-04576-f005:**
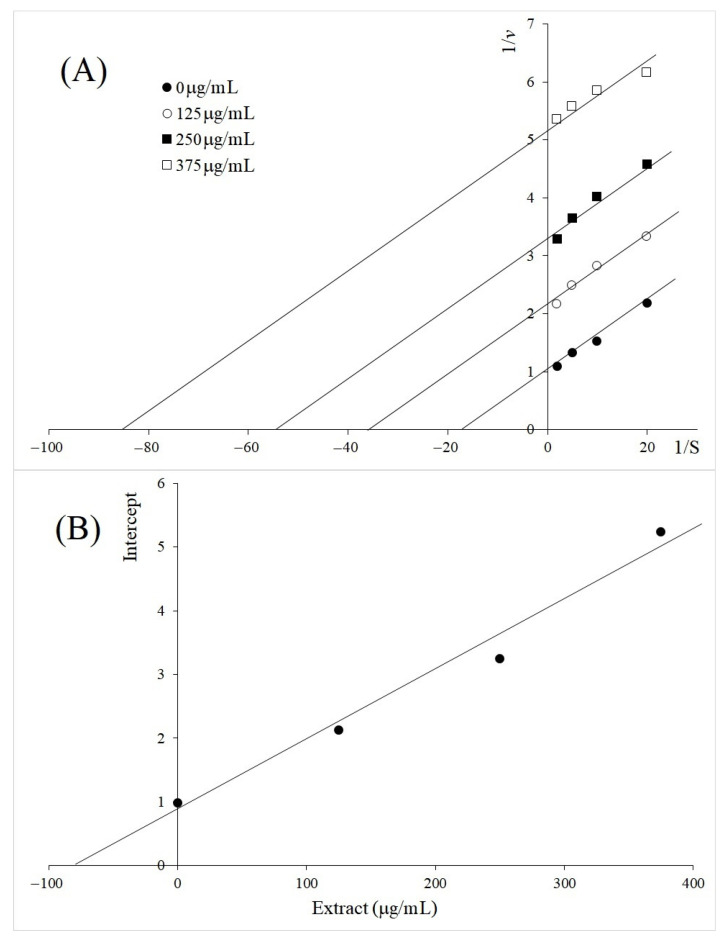
Inhibition kinetics of *M. oleifera* on TYR diphenolase activity at different concentrations (125, 250, and 375 µg/mL). (**A**) Lineweaver–Burk plot; (**B**) secondary plot between inhibitor concentrations and intercept (linear equation; y = 0.01112x + 0.8113, *R*^2^ = 0.9772).

**Figure 6 molecules-26-04576-f006:**
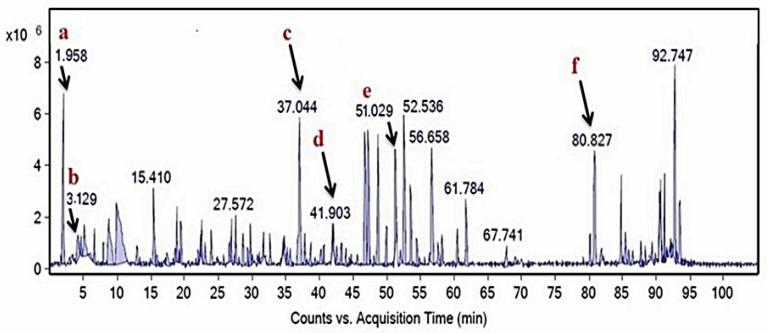
LC-QTOF MS chromatogram of *M. oleifera* ethanol extract. The identified compounds shown are: (**a**) lactobionic acid; (**b**) l-ascorbic acid-2-glucoside; (**c**) luteolin 7-(6‴-acetyl allosyl-(1->2)-glucoside); (**d**) luteolin 7-methyl glucuronide; (**e**) 3-hydroxylidocaine glucuronide; (**f**) luteolin.

**Figure 7 molecules-26-04576-f007:**
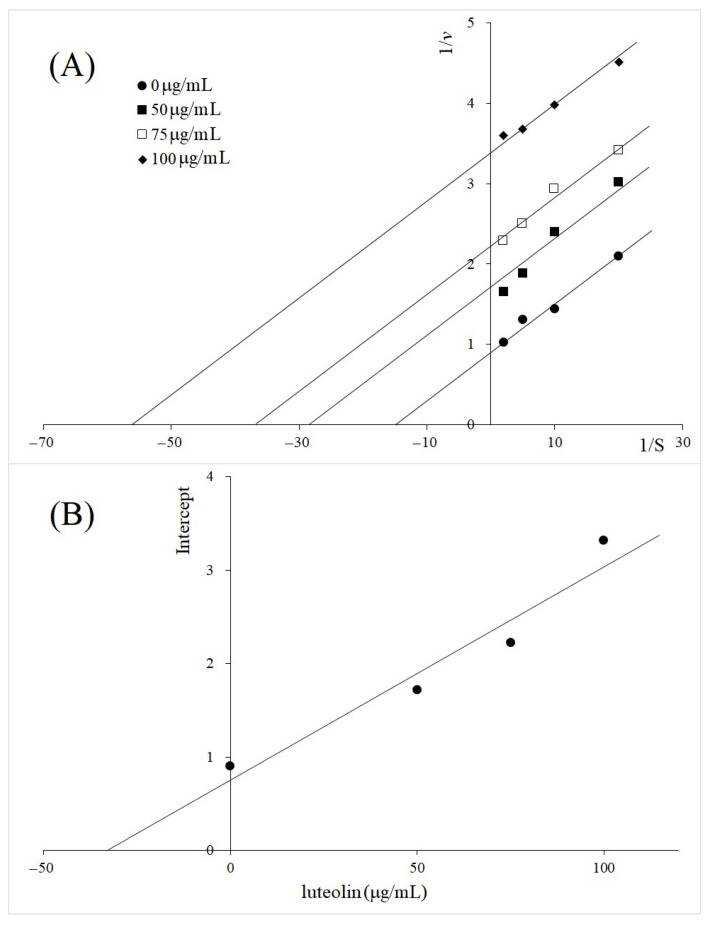
Tyrosinase inhibition kinetics of luteolin at the different concentrations of 50, 75 and 100 µg/mL. (**A**) Lineweaver–Burk plot; (**B**) secondary plot for the inhibitor concentration and intercept (linear equation; y = 0.02295x + 0.7491, *R*^2^ = 0.9373).

**Table 1 molecules-26-04576-t001:** Total phenolic content (TPC) and IC_50_ values for TYR.

Plant Extract	TPC (mg GAE/g Extract)	IC_50_ (µg/mL) of TYR
*Alpinia galanga* leaf	448.1 ± 2.3	240.4 ± 0.3
*Alpinia galanga* rhizome BL	651.9 ± 4.2	105.4 ± 0.2
*Alpinia galanga* rhizome YL	121.0 ± 0.3	294.6 ± 0.3
*Moringa oleifera* leaf	472.4 ± 2.8	121.3 ± 0.4
*Panax ginseng* leaf	146.5 ± 0.8	311.2 ± 0.4
*Panax ginseng* rhizome	233.0 ± 1.0	324.0 ± 0.5
*Vitis vinifera* seed	364.8 ± 1.2	94.8 ± 0.2
Ascorbic acid	- ^1^	235.7 ± 1.0
Kojic acid	- ^1^	192.3 ± 0.4

^1^ Not determined.

**Table 2 molecules-26-04576-t002:** Inhibition of TYR diphenolase by plant extracts.

Plant Extract	Type of Inhibition	Inhibition Constants, µg/mL	
		*K_i_*	*K_ii_*
*Alpinia galanga* rhizome BL	Mixed	200	128
*Vitis vinifera* leaf	Mixed	42	68
*Molinga oleifera* leaf	Uncompetitive		73
Luteolin	Uncompetitive		33

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
