# Peer review of "Alternative Approach for Specific Tyrosinase Inhibitor Screening: Uncompetitive Inhibition of Tyrosinase by *Moringa oleifera"

_molecules, 2021, doi:10.3390/molecules26154576_

Round 1
Reviewer 1 Report
Dear Molecules Editors:
Thank you for the opportunity to review the manuscript entitled “Alternative Approach for the Specific Tyrosinase Inhibitor Screening: Uncompetitive Inhibition of Tyrosinase by Moringa Oleifera” ( manuscript ID: molecules-1264850). While the subject matter of the manuscript is appropriate for the audience of Molecules, and the authors have clearly demonstrated experimental mastery of the techniques as present in the manuscript, it is not clear that the data supports the conclusion that luteolin is a specific inhibitor of mushroom tyrosinase. This is the fundamental conclusion of the manuscript as evidenced by the manuscript’s title. For the following reasons, I am recommending that the manuscript in its current state be rejected.
Luteolin is a very well characterized natural product that has been ascribed a host of various biological activities, with more than 153 referenced publications in Molecules alone (via a Molecules website search). This suggests that luteolin may likely fall into the category of Pan-Assay Interfering Substances (PAINS, Baell, JB., https://doi.org/10.1021/acs.jnatprod.5b00947 ). In order to gain insight into whether this is the case, I would recommend adding a nonionic detergent to all TYR reactions and determining what effect this has on the observed potency and mechanism of action for both the extracts themselves and the luteolin control. Similar studies have been carried out for luteolin in the context of DXR inhibition ( Bioorganic Chemistry,Volume 59,2015, Pages 140-144, https://doi.org/10.1016/j.bioorg.2015.02.008 ), demonstrating that luteolin is likely an aggregating compound and not a specific inhibitor. Without these (or equivalent experiments) to assess the specificity of Moringa Oleifera extracts, the data only supports that TYR is one of the dozens of characterized enzymes which show sensitivity to luteolin.
The authors allude to the potential that luteolin glucuronides and glycosides may also have TYR inhibitory capacity. If this is the case, then the inclusion of data supporting this claim would greatly strengthen this manuscript. As the authors rightly note, luteolin has already been identified as an uncompetitive TYR inhibitor (reference 19). This reduces the novelty of the authors findings and so expanding this mechanism of inhibition to luteolin derivatives would be desirable.
Secondarily to above revision requests, I would recommend adding error bars to the inhibition kinetics graphs as well as some indication of “goodness of fit” to the equation (r-squared or 95% confidence intervals). If it is possible to directly plot the kinetic data using curve fitting software with non-linear regression capabilities, this make the representation of the data more clear to the reader.
Reviewer 2 Report
The manuscript investigates the inhibition of tyrosinase by the extracts of different plants commonly used in Asian cuisine. Using a kinetics study, the authors found that one of the examined extracts exhibits uncompetitive inhibition and suggest that luteolin and its derivatives present in that plant extract are responsible for this feature. The introduction should be written in more detail: for example, the sentence at lines 95-96 mentions curcuminoids. In the following sentence, the authors are referring to the increased utilization of grape seeds. It would be nice if this connection (curcuminoids and grape seeds) is explicitly explained in the text. The main drawback of the manuscript is poor English, so I strongly suggest language editing.
Reviewer 3 Report
In the present work, Hashim and colleagues describe different TYR diphenolase inhibitors such as extracts from V. vinifera seed, A. galanga rhizome and M. oleifera leaf, that were selected from a total of five plant samples based on the inhibition kinetic studies. Among the selected plant extracts, the authors found that, M. oleifera leaf presents a rare uncompetitive inhibition pattern. Therefore, this plant extract was subjected to HPLC-MS analysis in order to identify the specific compounds responsible for the uncompetitive inhibition of TYR diphenolase.
Even though the novelty of the study does not consist of modern approaches, the results obtained may be of interest for different industries taking in account the fact that the identified bio-active compounds are from plants for consumption.
The article is well written and structured. I just have a few recommendations.
In order to present the results more clearly, I recommend to insert in a table all the data obtained through the HPLC-MS analysis. For example, the head of the table could contain the following information: the identified compounds, the retention time, the [M-H] (m/z), and the product ions (m/z).
Please, see the line 216 and remove the verb, “to interact”.
Reviewer 4 Report
- Page 2, Line 59; please give the definition and clear cutoff Km and Vmax for uncompetitive inhibition.
- Page 3. Table 1; Please explain how to calculate and correlate the TPC of the plants from mg GAE/g extract.
- Page 4. Figure 2; Before showing the correlation between TPC and IC50 TYR, I would suggest authors should demonstrate the HPLC chromatogram. To compare the secondary metabolites profile of the extracts and to prove the polyphenolics components in each plant.
- Page 8, Line 182; Authors proposed the glycosides of luteolin can be the active compounds. The HPLC-MS analysis can't be used as the solid supporting evidence, please survey and cite the references.
Round 2
Reviewer 1 Report
The authors have made a good faith attempt to address concerns raised in my initial review. While I would have preferred to see the error bars associated with individual data points included in graphs for Figures 3-5 and Figure 7, I appreciate that the authors did include a goodness of fit measure (r-squared) for their fitted lines. Along the same lines, the authors have included an additional experiment determining the effect of Triton X-100 on observed inhibition by luteolin and an M. oleifera extract, for which they are to be commended. While I would have preferred additional methods describing how this experiment was carried out (what substrate concentration was used for example), the results as presented suggest that in this context luteolin is not exhibiting aggregation.
Given the well annotated history of luteolin as a reported biochemical inhibitor of numerous other targets, doubts remain as to whether it can be claimed as a specific inhibitor of TYR (as stated in the article title); however, that is now an editorial rather than a reviewer concern.
Reviewer 4 Report
The reply is acceptable.